

# A 3-hour, 1-km surface soil moisture dataset for the contiguous United States from 2015 to 2023

Haoxuan Yang [a*], Jia Yang [a], Tyson E. Ochsner [b], Erik S. Krueger [b], Mengyuan Xu [c],

5   Chris B. Zou [a]

Affiliation

[a] Department of Natural Resource Ecology and Management, Oklahoma State University, Stillwater, OK 74078, USA.

10   [b] Department of Plant and Soil Sciences, Oklahoma State University, Stillwater, OK 74078, USA.

[c] Institute of Agricultural Science and Technology Information, Shanghai Academy of Agricultural Sciences, Shanghai 201403, China.

*Correspondence to*: Haoxuan Yang (yhx965665334@163.com)



**Abstract.**

Surface soil moisture (SSM) is a critical variable for understanding the terrestrial hydrologic cycle, and it influences ecosystem dynamics, agriculture productivity, and water resource management. Although SSM information is widely estimated through satellite-derived and model-assimilated methods, datasets with fine spatio-temporal resolutions remain unavailable at the continental scale, yet are essential for improving weather forecasting, optimizing precision irrigation, and enhancing fire risk assessment. In this study, we developed a new 3-hour, 1-km spatially seamless SSM dataset spanning 2015 to 2023, covering the entire contiguous United States (CONUS), using a spatio-temporal fusion model. This approach effectively combines the distinct advantages of two long-term SSM datasets, namely, the Soil Moisture Active Passive (SMAP) L4 SSM product and the Crop Condition and Soil Moisture Analytics (Crop-CASMA) dataset. The SMAP product provides spatially seamless SSM observations with a 3-hour temporal resolution but at a 9-km spatial resolution, while the Crop-CASMA SSM dataset offers a finer spatial resolution of 1 km but has a daily temporal resolution and contains spatial gaps. To overcome the spatio-temporal mismatch between the two products, we developed a time-series data mining approach known as the highly comparative time-series analysis (HCTSA) method to extract multiple spatially seamless characteristics (e.g., maximum and mean) from the two inter-annual SSM datasets (i.e., SMAP and Crop-CASMA). Then the fusion model was constructed using the extracted 9-km and 1-km characteristics and each scene of the SMAP, in turn. Finally, the 3-hour, 1-km SSM data (named as STF_SSM) were predicted from 2015 to 2023. The comparison with *in-situ* data from multiple SSM observation networks showed that the performance of our STF_SSM dataset is better than the Crop-CASMA and is close to the SMAP L4 product, with mean correlation coefficients (CC) of 0.716 at the daily scale and 0.689 at the 3-hour scale. The STF_SSM dataset in this study is the first long time-series, spatially seamless SSM dataset to realize continuous intra-day 1-km SSM observations every 3 hours across the CONUS, which provides a new insight into the fast changes in soil moisture along with drought and wet spell occurrences, and ecosystem responses. Additionally, this dataset provides a valuable data source for the calibration and validation of land surface models. The STF_SSM dataset is available at https://doi.org/10.6084/m9.figshare.28188011 (Yang et al., 2025).





## 1. Introduction

Surface soil moisture (SSM) is an important component of global hydrological cycling and

serves as a key indicator of drought occurrences **(Souza et al., 2021; Krueger et al., 2024)**, climate change **(Guillod et al., 2015)**, and ecosystem functions **(Green et al., 2019; Liu et al., 2020a)**. To better understand the spatio-temporal changes in SSM, the Soil Moisture Active Passive (SMAP) satellite, launched in 2015, provides SSM data on a global scale at spatial resolutions of 9-km and 36-km using the onboard L-band radiometer **(Entekhabi et al., 2010;**

**Chan et al., 2016)**. SMAP products are generated at four levels of processing. Retrieved from brightness temperature information observed by satellites, SMAP Level 2 (L2) and Level 3 (L3) SSM data are half-orbital and daily composited. However, considering the non-overlapping revisit orbit and snow coverage, spatial gaps are inevitable in SMAP L2 and L3 SSM products. To solve this problem, SMAP Level 4 (L4) SSM product assimilates SMAP L2

and L3 data into a land surface model and provides spatially seamless SSM and root zone soil moisture estimates with a temporal resolution of 3 hours and spatial resolution of 9 km.

Numerous studies have investigated the performance of satellite-derived SSM based on the triple collocation analysis **(Chen et al., 2018)**, information theory **(Kumar et al., 2018)**, and ground-based *in-situ* data **(Kim et al., 2018)**. Results showed that the SMAP data typically

have better performance at a global scale compared with other satellite-derived SSM products, such as the Soil Moisture and Ocean Salinity (SMOS), the Advanced Scatterometer (ASCAT), and Advanced Microwave Scanning Radiometer 2 (AMSR2). For example, **Ma et al. (2019)** and **Min et al. (2023)** demonstrated that SMAP data outperformed the SMOS, AMSR2, and the Climate Change Initiative of the European Space Agency (CCI) SSM product in terms of

capturing temporal dynamics. **Montzka et al. (2017)** evaluated different SSM products in six regions and found that the SMAP data had greater accuracy than SMOS, AMSR2, and ASCAT. Additionally, SMAP data outperformed other satellite-derived SSM products in the Gilgel Abay watershed of Ethiopia **(Alaminie et al., 2024)**, the Genhe area in China **(Cui et al., 2017)**, and the Huai River basin of China **(Wang et al., 2021)**.

Although the SMAP data are stable and reliable, potential applications are constrained by their coarse spatial resolution. To address this issue, several downscaling strategies have been

applied by integrating optical/thermal-infrared data **(Peng et al., 2017; Sabaghy et al., 2020; Abbaszadeh et al., 2021; Meng et al., 2024)**. In general, the universal triangle feature **(Carlson et al., 1990, 1995)** and trapezoidal feature spaces **(Moran et al., 1994; Merlin et al.,**

**2012)** provide the theoretical basis for most downscaling studies. The universal triangle feature primarily leverages land surface temperature (LST) and normalized difference vegetation index (NDVI) from optical/thermal-infrared data, e.g., the Moderate Resolution Imaging Spectroradiometer (MODIS), to capture spatio-temporal variations in SSM, highlighting that SSM is closely related with LST and NDVI. **(Carlson, 2007)**. Compared to the universal

triangle feature, the trapezoidal feature considers the influence of the fraction of water-stressed vegetation **(Djamai et al., 2016)**. By incorporating fine-resolution LST and NDVI at various fractional vegetation cover conditions, the effects of evaporation (e.g., soil evaporative efficiency) can be quantified, enabling the development of SSM products at high spatial resolution **(Merlin et al., 2012; Kim and Hogue, 2012; Molero et al., 2016)**. Therefore, fine-

resolution LST and NDVI are often employed as auxiliary data for SSM downscaling.

Commonly, downscaling approaches are based on geostatistical models, which consider the spatial variations of LST and NDVI within and outside of the SSM pixel **(Song et al., 2019)**. For example, a regression kriging-based model and its modified version were used to disaggregate the coarse pixels in SSM data using LST and NDVI data **(Jin et al., 2018; Wen**

**et al., 2020; Jin et al., 2021; Yang et al., 2024)**. Robust ensemble learning approaches have also been used to downscale SSM **(Zhao et al., 2018; Wei et al., 2019; Karthikeyan and Mishra, 2021)**. For example, by integrating multiple decision tree models, a random forest model was employed to downscale SMAP SSM data from 36 km to 1 km **(Hu et al., 2020)**. This approach can include fine-resolution auxiliary information, such as topography, location,

and soil texture **(Abbaszadeh et al., 2019; Liu et al., 2020b; Guevara et al., 2021; Wang et al., 2022)**. Deep learning is another popular downscaling approach, as the strong fitting capability effectively characterizes the SSM using LST, NDVI, and other auxiliary information **(Xu et al., 2022; Zhao et al., 2022; Xu et al., 2024)**.

Using the aforementioned methods, many high-resolution SSM datasets have been

developed **(Han et al., 2023; Brocca et al., 2024)**. However, the optical/thermal-infrared auxiliary data are usually disturbed by atmospheric conditions, such as clouds and haze **(Ma et**



**al., 2022a, b)**, resulting in difficult disaggregation of coarse SSM pixels under clouds or haze. To mitigate these issues, multiple-day composited optical/thermal-infrared data are often used as the auxiliary variables for producing the SSM dataset **(Li et al., 2022; Zheng et al., 2023)**.

Additionally, reconstruction of the missing optical/thermal-infrared data is a reliable choice, which is then used for SSM generation **(Long et al., 2019; Abowarda et al., 2021; Song et al., 2022)**. For example, **Zhao et al. (2021)** reconstructed the seamless LST data and used them to generate the SSM dataset at a 1-km spatial resolution. In addition, integrating other high-resolution SSM products is also an appropriate method, which can avoid the influence of the

optical/thermal-infrared auxiliary data **(Jiang et al., 2019; Yang et al., 2022; Jiang et al., 2024)**.

High-resolution SSM datasets have been developed based on the original SMAP SSM product. For instance, the National Aeronautics and Space Administration (NASA) combined data from the Sentinel-1 satellites' synthetic aperture radar with SMAP's passive radiometer

to produce the SSM product (SPL2SMAP_S), which offers a spatial resolution of 3 km **(Jagdhuber et al., 2019)**. However, differences in the revisit orbits of SMAP and Sentinel-1, coupled with the narrower swath width of Sentinel-1 compared to SMAP, restrict the spatial coverage of the SPL2SMAP_S product **(Das et al., 2019; Kim et al., 2021)**. In addition, the United States Department of Agriculture's National Agricultural Statistics Service (USDA-

NASS) has developed a daily, 1-km resolution SSM product within the Crop Condition and Soil Moisture Analytics (Crop-CASMA) system **(Colliander et al., 2019; Zhang et al., 2022)**. This product disaggregates the 9-km satellite-derived SMAP SSM data by incorporating auxiliary 1-km data from MODIS **(Liu et al., 2021, 2022)**. Although the Crop-CASMA SSM product provides sufficient spatial details, it retains spatial gaps inherent to the daily SMAP

SSM data, limiting its overall spatial continuity. Similarly, the lack of spatial information also restricts the application of other high-resolution SSM datasets **(Fang et al., 2022; Lakshmi and Fang, 2023; Yang et al., 2024).**

Finer spatial and temporal resolution have become increasingly important for SSM datasets to facilitate more accurate monitoring of dynamic soil moisture variations. For long time-series

and large-scale SSM datasets, a 1-km spatial resolution is commonly adopted, as daily auxiliary data at 1-km resolution can be extracted from MODIS. However, the commonly used 1-km

SSM datasets at a large scale have a daily or even coarser temporal resolution, limiting their capacity to depict the intra-day SSM variations. This highlights the challenge of achieving both high temporal and spatial resolution in SSM datasets simultaneously.

In this work, we generated a 3-hour, 1-km SSM dataset (denoted as STF_SSM) for the contiguous United States (CONUS) from 2015 to 2023. Based on an advanced and efficient spatio-temporal fusion model, the advantages of high observation frequency (3-hour) in the SMAP L4 SSM product and satisfactory spatial details (1-km) in the Crop-CASMA SSM dataset were integrated into a new SSM dataset. A time-series data mining approach was

employed to extract multiple spatially seamless characteristics from the SMAP L4 and Crop-CASMA SSM datasets, effectively addressing the spatio-temporal mismatches between the two input datasets within the fusion model. To evaluate the performance of the STF_SSM dataset, ground-based *in-situ* measurements were used for validation at both 3-hour and daily scales. The generated STF_SSM dataset facilitates intra-day SSM observations, providing a

valuable resource for the related studies.



## 2. Data and methods

### 2.1 Data

#### 2.1.1 SMAP L4 SSM product

SMAP L4 SSM data were downloaded from https://nsidc.org. The temporal and spatial resolutions of the SMAP L4 SSM product are 3-hour and 9-km, respectively. The SMAP L4 SSM product has a spatially complete coverage at a global scale. Validation studies showed that the SMAP L4 product provides more accurate and stable performance than SMAP L3 across all seasons **(Tavakol et al., 2019)**. In this work, we used the latest version 7 SMAP L4

geophysical dataset.

#### 2.1.2 Crop-CASMA SSM data

    The Crop-CASMA system integrates crucial vegetation and soil moisture data for the CONUS (such as SSM, root-zone soil moisture, and NDVI). These data are continuously updated and can be freely accessed from the USDA-NASS website at

https://nassgeo.csiss.gmu.edu/CropCASMA/. The system supports direct download, analysis, and visualization. In this study, the Crop-CASMA SSM data are derived from the SMAP Thermal Hydraulic disaggregation of Soil Moisture (SMAP THySM) dataset, which can provide 1-km daily SSM data and have two days of latency **(Liu et al., 2021, 2022; Zhang et al., 2022)**.

#### 2.1.3 *In-situ* data

    *In-situ* data are measured and recorded by ground-based sensors at different depths, which has often been used as the reference for validation of satellite-derived SSM datasets **(Dorigo et al., 2015)**. In this study, *in-situ* data from 2015 to 2023 were obtained from the International Soil Moisture Network (https://ismn.earth/en/) and the Oklahoma Mesonet

(https://www.mesonet.org/), with measurements taken at a depth of 5 cm **(McPherson et al., 2007)**. Noted that the available *in-situ* data were further filtered to ensure that only one site was included per SSM pixel. Detailed information and locations of the *in-situ* networks were presented in Table 1 and Figure 1.

**Table 1.** Details of the selected *in-situ* data and soil moisture observation network for validation.

| Network | Site number | Sensor |
|---|---|---|
| ARM | 14 | Hydraprobe II Sdi-12 E |
| CW3E | 16 | CS616/ Stevens-Hydra-Probe |
| FLUXNET-AMERIFLUX | 2 | CS655/ ThetaProbe-ML2X |
| MESONET | 123 | Campbell Scientific 229-L |
| PBO_H2O | 130 | GPS |
| SCAN | 183 | Hydraprobe-Analog-(2.5-Volt)/ Hydraprobe-Digital-Sdi-12-(2.5-Volt) |
| SNOTEL | 261 | Hydraprobe-Analog-(2.5-Volt)/ Hydraprobe-Analog-(5.0-Volt) |
| TxSON | 15 | CS655 |
| USCRN | 91 | Stevens-Hydraprobe-II-Sdi-12 |

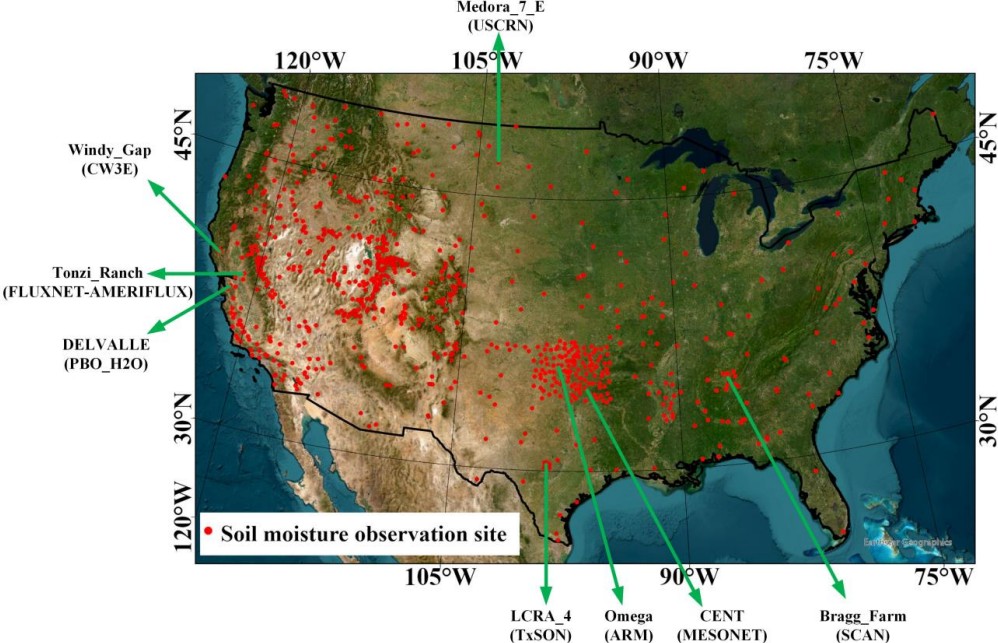

**Figure 1.** Spatial distribution of the *in-situ* soil moisture observation sites used in this study
for soil moisture validation. Each red point refers to one site. The eight marked sites show temporal variations of Surface Soil Moisture (SSM) in Figures 6 and 7, respectively. The basemap is from Esri, Earthstar Geographics, and the GIS User Community.



### 2.1.4 Land cover and terrain data

In this study, we validated the performance of our developed datasets for different land
cover types and different terrain conditions. Land cover type data were from the National Land
Cover Database (NLCD) product available at https://www.mrlc.gov/, which includes annual
land cover type at a spatial resolution of 30 m **(Homer et al., 2020; Jin et al., 2023)**.
Additionally, 30-m digital elevation models (DEM) from the NASA Shuttle Radar Topography
Mission project were utilized to describe the terrain information in the CONUS **(Rabus et al.,**
**2003)**.

### 2.2 Method

### 2.2.1 Characteristic extraction

Because of the spatial gaps in the Crop-CASMA SSM data, it is difficult to directly exploit
the daily Crop-CASMA SSM scene for the construction of the spatio-temporal fusion model
(see next section). To deal with this problem, a time-series mining approach, i.e., highly
comparative time-series analysis (HCTSA), was adopted to extract four spatially seamless 1-
km characteristics (including maximum, minimum, mean and median) in each pixel from the
Crop-CASMA SSM time-series data **(Fulcher et al., 2013; Fulcher and Jones, 2017)**.
Additionally, to match the extracted characteristics from the Crop-CASMA SSM data, the
same characteristics were also extracted at 9-km resolution from the corresponding SMAP L4
time-series data.

Even though the location of spatial gaps in the Crop-CASMA SSM time-series are varying
over time, the extracted HCTSA-based characteristics are not affected **(Yang and Wang, 2023).**
Typically, some factors that significantly influence SSM (e.g., precipitation, vegetation, and
temperature) exhibit periodic changes in the year, indicating that inter-annual fluctuations in
SSM tend to follow a periodic pattern. Thus, we selected a one-year temporal span for
extracting these characteristics of maximum, minimum, mean, and median SSM. The extracted
characteristics were then utilized to generate the corresponding STF_SSM scene.

### 2.2.2 Spatio-temporal fusion model

In this study, the virtual image pair-based spatio-temporal fusion (VIPSTF) model was



employed to generate the 3-hour, 1-km STF_SSM dataset, due to its stable performance, superior computational efficiency, and flexible usage **(Wang et al., 2020; Yang et al., 2023)**. The operation of the VIPSTF model requires at least one or multiple known image pairs at different spatial resolutions (a data pair is defined as one coarse and one fine resolution

characteristic extracted from the same year). Here, the extracted 1-km and 9-km HCTSA-based characteristics (i.e., maximum, minimum, mean, and median of SSM time-series) from the SMAP L4 and Crop-CASMA SSM time-series (i.e., four image pairs) were blended using the VIPSTF model. Specifically, each STF_SSM scene is produced as follows:

$$\mathbf{\widehat{STF\_SSM}}_t = \mathbf{STF\_SSM}_{VIP} + \mathbf{\Delta STF\_SSM}_t \ , \tag{1}$$

where $\mathbf{\widehat{STF\_SSM}}_t$ is the generated 3-hour, 1-km SSM scene at time $t$, $\mathbf{\Delta STF\_SSM}_t$ refers to

the increment data of the model at a 1-km spatial resolution at time $t$. The virtual $\mathbf{STF\_SSM}_{VIP}$ scene was predicted using a linear combination of the four extracted 1-km characteristics from the Crop-CASMA SSM time-series data in Eq. (2):

$$\mathbf{STF\_SSM}_{VIP} = \sum_{i=1}^{n} a_i \mathbf{F\_C}_i + b \ , \tag{2}$$

where $a_i$ is the coefficient for the $i$-th extracted fine characteristic $\mathbf{F\_C}_i$ at a 1-km spatial resolution. $n$ is the number of the characteristics ($n = 4$ in this study), and $b$ denotes a constant.

Based on the assumption of scale invariance **(Wang and Atkinson, 2018)**, the optimal coefficients $a_i$ and $b$ for each 3-hour SMAP scene were calculated in a linear regression, as follows:

$$\mathbf{SMAP}_t = \sum_{i=1}^{n} a_i \mathbf{C\_C}_i + b + \mathbf{\Delta SMAP}_t \ , \tag{3}$$

where $\mathbf{SMAP}_t$ refers to the known SMAP L4 SSM scene at time $t$, and $\mathbf{C\_C}_i$ is the $i$-th extracted coarse characteristic at a 9-km spatial resolution from the SMAP L4 SSM time-series

data. $\mathbf{\Delta SMAP}_t$ represents the residual data from the regression at time $t$. Moreover, the increment data (i.e., $\mathbf{\Delta STF\_SSM}_t$) in Eq. (1) can be disaggregated by the following spatial weighting scheme:

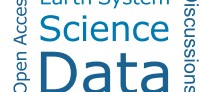

$$\Delta\textbf{STF\_SSM}_t(x_0,y_0)= \sum_{k=1}^{s} w_j \Delta\textbf{SMAP}_t(x_k,y_k) \,, \tag{4}$$

In Eq. (4), $(x_k,y_k)$ denotes the spatial distribution of the $k$-th similar pixel surrounding the center pixel $(x_0,y_0)$. The number of surrounding similar pixels is represented by $s$. Additionally, $w_j$ represents a weight calculated based on the distance between the center pixel and the $k$-th surrounding similar pixels. $\Delta\textbf{SMAP}_t$ was interpolated from 9 km to 1 km using the bicubic method.

### 2.2.3 Data generation

Based on the VIPSTF model, the 3-hour, 1-km STF_SSM dataset was generated from 2015-04-01 to 2023-12-31. The generation flowchart is depicted in Figure 2. The specific process steps for producing the STF_SSM dataset are described as follows:

(1) For each year, 1-km and 9-km spatially seamless characteristics (i.e., maximum, minimum, mean, and median) were extracted from the Crop-CASMA and SMAP L4 SSM time-series using the HCTSA method, respectively.

(2) To generate a STF_SSM scene at time $t$, a VIPSTF model was constructed using the extracted characteristics in step (1) and a SMAP L4 SSM scene at time $t$ within the year. Then, the STF_SSM scene at time $t$ was generated.

(3) The aforementioned steps were repeated each 3-hour period to produce the 3-hour, 1-km STF_SSM data.

Finally, a total of 25,567 STF_SSM scenes were produced. The Pete High-Performance Computing (HPC) facility at Oklahoma State University was employed for data generation.



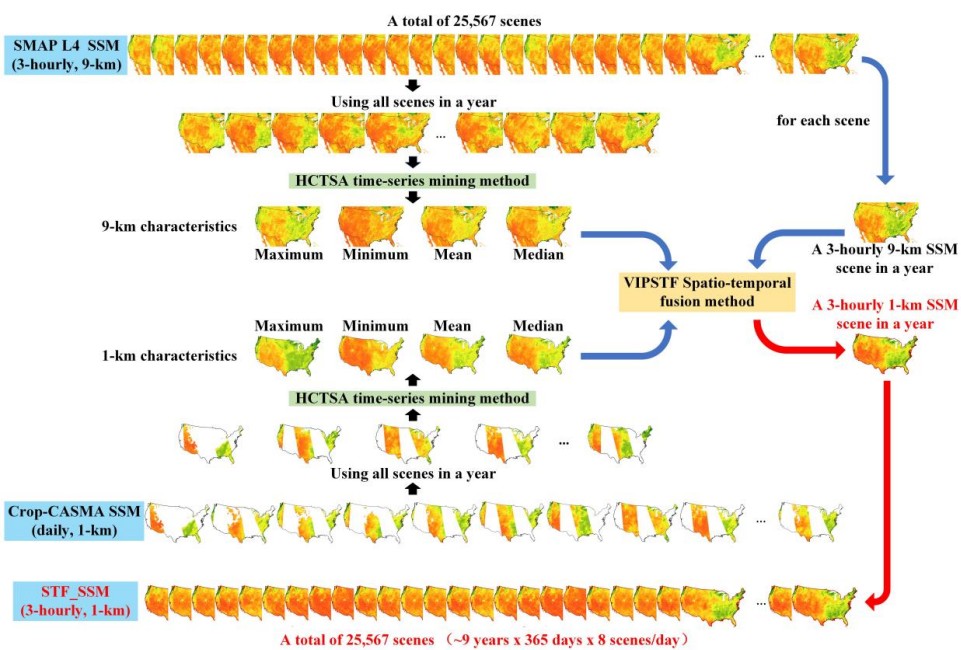

**Figure 2.** Flowchart for generating the 3-hour, 1-km STF_SSM dataset from 2015 to 2023.

**2.3 Validation**

In this paper, we divided the soil moisture observations (Figure 1) into two groups (3-hour group and daily group) for the validation of our generated data. Considering that the time of SMAP L4 and STF_SSM data is not on the hour (at 1:30, 4:30, 7:30, 10:30, 13:30, 16:30, 19:30, and 22:30), the average SSM value between two adjacent integer times was used to represent the SSM value at the specific time. For example, when validating the SMAP L4 and STF_SSM

scenes at 4:30, the mean value of the 4:00 and 5:00 *in-situ* data was used to represent the soil moisture at 4:30. For daily scale validation, the hourly *in-situ* data were averaged to obtain daily values. Similarly, the 3-hour SMAP L4 and STF_SSM data were composited into daily scenes. To assess accuracy, five widely used statistical metrics were adopted, that is, the correlation coefficient (CC), root mean square error (RMSE), bias (Bias), unbiased root mean

square error (ubRMSE), and Kling-Gutpa efficiency (KGE).

## 3. Results

### 3.1 Spatial pattern of the developed SSM dataset

The spatial pattern of the Crop-CASMA SSM, the SMAP L4 SSM, and the generated
STF_SSM datasets are shown in Figure 3 at four randomly selected time points (i.e., 2015-04-01, 2017-06-08, 2019-08-16, and 2021-10-25). The 1-km Crop-CASMA SSM dataset has
spatial gaps and does not have wall-to-wall data covering the entire CONUS. In contrast, both
the SMAP L4 and STF_SSM datasets can provide spatially seamless observations. It is noted
that the SMAP L4 SSM scenes contain some abnormal pixels with extremely high SSM values
(SSM values of 0.6 and higher), especially in the northern part of the CONUS (e.g., some pixels
around the Great Lakes in Figure 3).

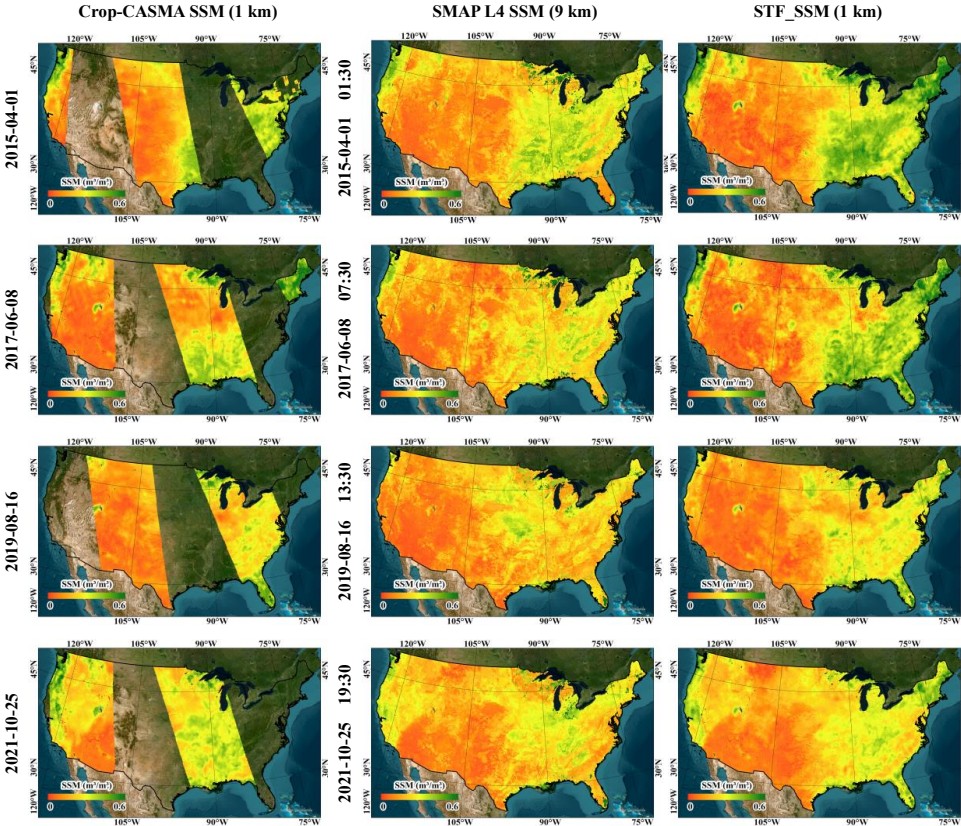

**Figure 3.** Spatial pattern of Surface Soil Moisture (SSM) in the Crop-CASMA SSM dataset
(left), SMAP L4 SSM product (middle), and the STF_SSM dataset (right) at four randomly
selected time points. Both the SMAP L4 and STF_SSM datasets are exhibited at the 3-hour





scale, while the Crop-CASMA SSM dataset is displayed at the daily scale. The basemap is from Esri, Earthstar Geographics, and the GIS User Community.

To illustrate the advantages of the STF_SSM dataset, we zoomed into a sub-region in the CONUS at a date with rainfall (2018-08-14) and showed SSM in the three different datasets in Figure 4. It is clear that both the 3-hour SMAP L4 SSM and STF_SSM datasets can capture

increased SSM values from 1:30 to 7:30 in the southwest region of the sub-region. Moreover, The STF_SSM and Crop-CASMA SSM datasets provide more detailed spatial information than those in the SMAP L4 SSM product. The spatial texture of the STF_SSM dataset closely resembles that of the 1-km Crop-CASMA SSM dataset, which is smoother than that of the SMAP L4 SSM product.

Next, we selected two random pixels in Nebraska (shown in Figure 4) to exhibit intra-day SSM variation (Figure 5). Although the Crop-CASMA SSM scene provides spatial information at a 1-km resolution, it only provides SSM value at the daily scale. In contrast, both the SMAP L4 and STF_SSM datasets show the changes in SSM every 3 hours. Furthermore, the changing patterns of SSM over time are similar between the SMAP L4 and STF_SSM datasets (the CC

values in Figure 5a and Figure 5b are 0.997 and 0.999), indicating the stability and consistency of the STF_SSM dataset.

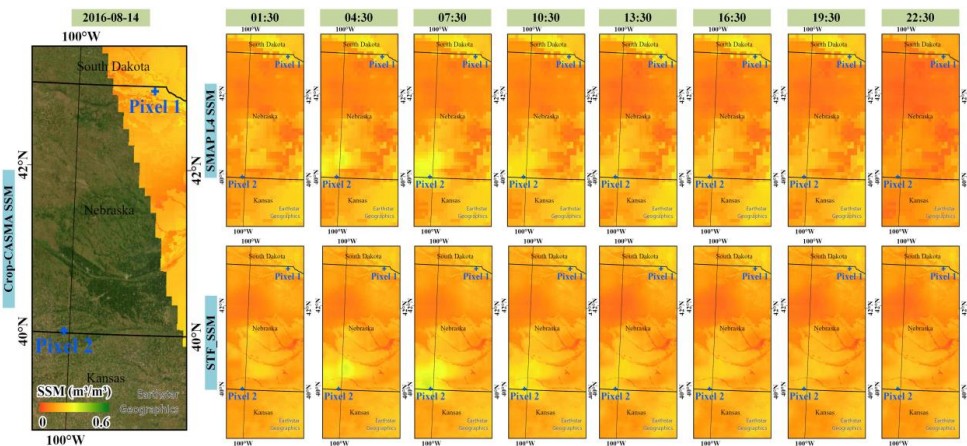

**Figure 4.** A sub-region in the western Continental United States (CONUS) to exhibit the spatial textures of the three Surface Soil Moisture (SSM) datasets on 2018-08-14. The basemap is
from Esri, Earthstar Geographics, and the GIS User Community.





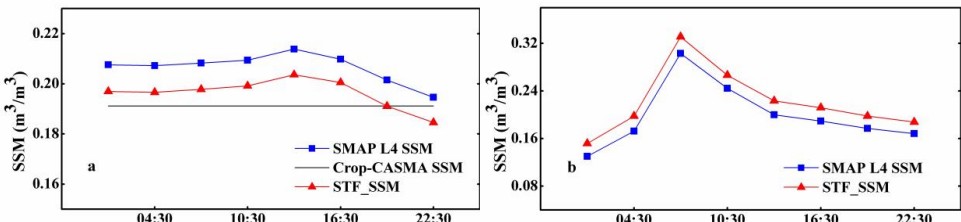

**Figure 5.** The intra-day Surface Soil Moisture (SSM) variation for two randomly selected pixels on 2018-08-14. The Crop-CASMA SSM dataset does not provide the intra-day SSM variation, shown as the flat black line. (a) and (b) refer to the pixels 1 and 2 in Figure 4, respectively. Since pixel 2 is located in the spatial gaps of the Crop-CASMA SSM scene, it does not exhibit the flat black line in (b).

### 3.2 Validation based on daily soil moisture observations

At the daily scale, the three daily SSM datasets (the Crop-CASMA SSM, SMAP L4 SSM, and STF_SSM datasets) were compared against the daily *in-situ* data from 9 soil moisture observation networks. Figure 6 shows the SSM time-series acquired from four randomly selected sites distributed across the CONUS: the Windy_Gap site in the CW3E network, CENT site in the MESONET network, DELVALLE site in the PBO_H2O network, and LCRA_4 site in the TxSON network. Although there are gaps in the *in-situ* data at these sites, the available *in-situ* data are sufficient for the validation of these three SSM datasets. Moreover, the SSM daily variations of the three SSM datasets are similar to those of the *in-situ* data. For example, the CC values for the SMAP L4, Crop-CASMA, and STF_SSM datasets in Figure 6a (Windy_Gap site in the CW3E network) are 0.924, 0.879, and 0.886, respectively. In addition, there are some biases between different SSM datasets due to differences in spatial resolution and derived methods. Specifically, the SMAP records a lower minimum SSM value at the Windy_Gap site and a higher SSM maximum at the Delvalle site, compared with the Crop-CASMA and STF_SSM datasets.

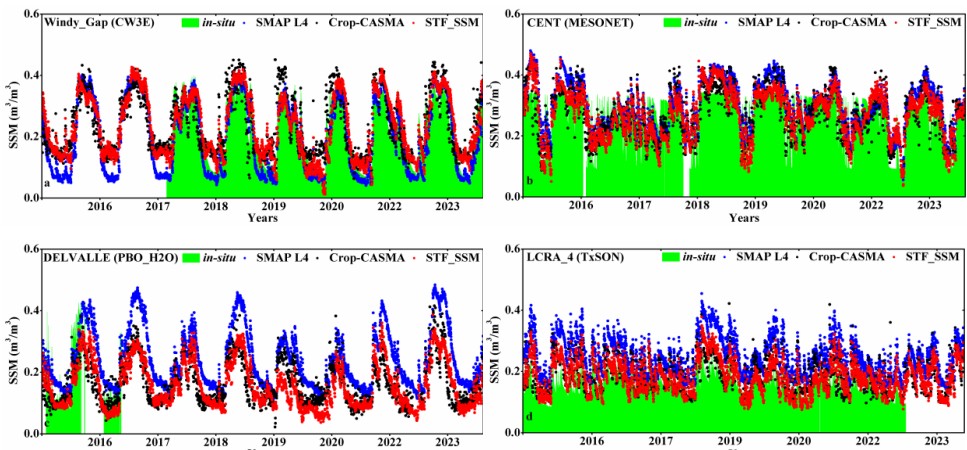

**Figure 6.** Temporal variations of daily surface soil moisture (SSM) from the SMAP L4 (blue),
Crop-CASMA (black), STF_SSM (red), and *in-situ* observation data (green) at four different
sites. (a) Windy_Gap site in the CW3E network. (b) CENT site in the MESONETX network.
(c) DELVALLE site in the PBO_H2O network. (d) LCRA_4 site in the TxSON network.





**Table 2.** Accuracy of the daily surface soil moisture (SSM) from the SMAP L4, the Crop-CASMA and the STF_SSM datasets. Values in bold indicate the dataset with the best performance for each statistic in each row. Underlined values indicate second best.

| | SMAP L4 SSM (9 km) | | | | | Crop-CASMA SSM (1 km) | | | | | STF_SSM (1 km) | | | | |
|---|---|---|---|---|---|---|---|---|---|---|---|---|---|---|---|
| | CC | RMSE | Bias | ubRMSE | KGE | CC | RMSE | Bias | ubRMSE | KGE | CC | RMSE | Bias | ubRMSE | KGE |
| ARM | **0.752** | 0.089 | -0.001 | **0.058** | 0.538 | 0.712 | 0.086 | 0.030 | 0.069 | 0.589 | 0.710 | **0.078** | 0.023 | 0.062 | **0.595** |
| CW3E | **0.859** | 0.070 | -0.012 | **0.042** | **0.408** | 0.531 | 0.111 | -0.046 | 0.078 | -0.058 | 0.818 | 0.093 | -0.053 | 0.047 | 0.181 |
| FLUXNET-AMERIFLUX | **0.925** | 0.055 | -0.012 | 0.051 | **0.758** | 0.875 | 0.069 | -0.028 | 0.062 | 0.740 | 0.892 | 0.070 | -0.034 | 0.061 | 0.645 |
| MESONET | 0.721 | 0.096 | 0.042 | 0.064 | 0.509 | 0.633 | 0.102 | 0.046 | 0.077 | 0.507 | 0.710 | **0.089** | 0.035 | **0.064** | **0.525** |
| PBO_H2O | 0.724 | 0.063 | 0.002 | 0.050 | 0.415 | 0.614 | 0.066 | 0.020 | 0.055 | 0.404 | 0.704 | **0.061** | 0.018 | **0.050** | **0.445** |
| SCAN | **0.660** | 0.087 | -0.012 | **0.056** | 0.191 | 0.470 | 0.102 | -0.012 | 0.078 | 0.007 | 0.621 | 0.089 | -0.019 | 0.057 | 0.155 |
| SNOTEL | **0.622** | 0.102 | 0.027 | **0.074** | 0.254 | 0.335 | 0.120 | 0.036 | 0.093 | 0.091 | 0.561 | 0.107 | 0.031 | 0.078 | 0.202 |
| TxSON | **0.771** | 0.085 | -0.019 | 0.045 | 0.361 | 0.598 | 0.079 | 0.025 | 0.059 | 0.340 | 0.751 | **0.065** | 0.018 | **0.043** | **0.521** |
| USCRN | **0.709** | 0.084 | -0.017 | **0.052** | 0.357 | 0.488 | 0.109 | -0.025 | 0.081 | 0.166 | 0.675 | 0.097 | -0.038 | 0.054 | 0.282 |
| Mean | **0.749** | 0.081 | -0.001 | **0.055** | 0.421 | 0.584 | 0.094 | 0.005 | 0.072 | 0.310 | 0.716 | **0.083** | -0.002 | 0.057 | 0.395 |


At the network level, the results of the accuracy assessment (Table 2) show that the 9-km

SMAP L4 SSM product has the greatest accuracy among the three datasets. At a spatial resolution of 1 km, the generated STF_SSM dataset outperforms the Crop-CASMA SSM dataset. Specifically, the mean CC for the SMAP L4 SSM product is 0.749, which is 0.033 and 0.165 higher than the STF_SSM and Crop-CASMA datasets, respectively. The RMSE and ubRMSE for the SMAP L4 SSM dataset are 0.081 $m^3/m^3$ and 0.055 $m^3/m^3$, which are the

smallest among the three datasets. The STF_SSM dataset has RMSE and ubRMSE values of 0.083 $m^3/m^3$ and 0.057 $m^3/m^3$, which is slightly higher than those for the SMAP L4 SSM product but smaller than those for the Crop-CASMA dataset. The Bias for the SMAP L4 SSM product (with a Bias of -0.001 $m^3/m^3$) and the STF_SSM dataset (with a Bias of -0.002 $m^3/m^3$) are closer to SSM observation data than that of the Crop-CASMA SSM dataset. The STF_SSM

dataset has a mean KGE of 0.395, which is 0.026 smaller than that of the SMAP L4 but 0.185 larger than that of the Crop-CASMA dataset. At the three SSM observation networks of MESONET, PBO_H2O, and TxSON, the STF_SSM dataset demonstrates better performance compared to the other datasets. However, for the other networks, the SMAP L4 SSM product outperforms the other datasets in terms of accuracy.

**3.3 Validation based on 3-hour soil moisture observations**

The 9-km SMAP L4 product and the 1-km STF_SSM dataset were compared with the 3-hour *in-situ* data from 7 networks. We randomly selected 4 sites (Omega site in the ARM network, Tonzi_Ranch site in the FLUXNET-AMERIFLUX network, Bragg_Farm site in the SCAN network, and Medora_7_E site in the USCRN network) to exhibit the SSM comparison

(Figure 7). The temporal variations of SSM time-series from the *in-situ* data, SMAP L4 product, and STF_SSM dataset are similar to each other, e.g., the CC values of the SMAP L4 and STF_SSM datasets in Figure 7a are 0.749 and 0.745 by referring to the *in-situ* data, revealing that both the SSM datasets can well capture the dynamics of SSM at the 3-hour scale. Moreover, the difference between the 9-km SMAP L4 product and 1-km STF_SSM dataset is small,

indicating that the downscaling of our STF_SSM dataset does not introduce significant errors in SMAP L4 SSM. Therefore, our STF_SSM dataset can be regarded as a reliable high-resolution version of the SMAP L4 product.

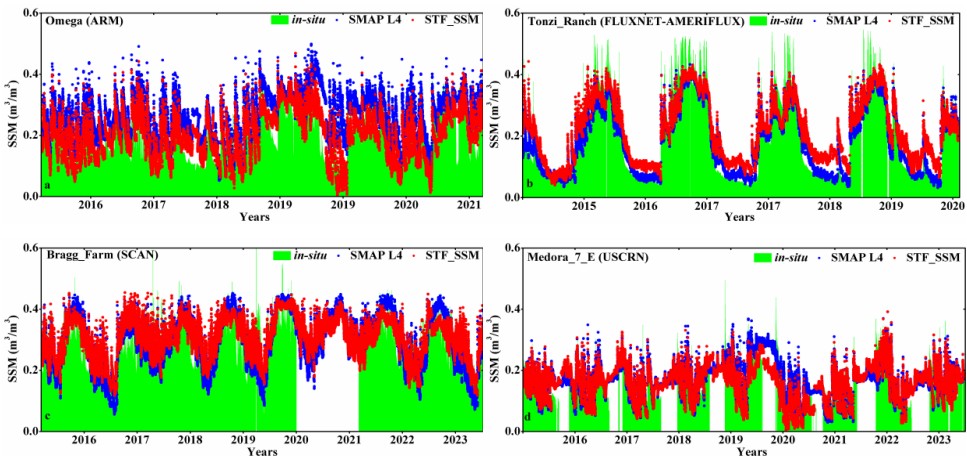

**Figure 7.** Temporal variations of 3-hour surface soil moisture (SSM) from SMAP L4 (blue), STF_SSM (red), and *in-situ* observation data (green) at four different sites. (a) Omega site in the ARM network. (b) Tonzi_Ranch site in the FLUXNET-AMERIFLUX network. (c) Bragg_Farm site in the SCAN network. (d) Medora_7_E site in the USCRN network.

The quantitative statistical metrics based on the 3-hour *in-situ* data are listed in Table 3. The results indicate that the 9-km SMAP L4 SSM product has better accuracy than the 1-km STF_SSM dataset. Specifically, the mean CC and KGE for the SMAP L4 SSM product are 0.728 and 0.414, which are 0.039 and 0.028 higher than those for the STF_SSM dataset. Furthermore, the SMAP L4 SSM product has a mean RMSE and ubRMSE of 0.086 $m^3/m^3$ and 0.059 $m^3/m^3$. Additionally, the Bias of the SMAP L4 SSM product is -0.005 $m^3/m^3$, which is closer to the SSM observation data than that of the STF_SSM dataset. For specific networks, we found that the CC and KGE of the SMAP L4 SSM dataset at the FLUXNET-AMERIFLUX network are 0.920 and 0.889, which are the highest among the 7 networks. On the other hand, the STF_SSM dataset provides an RMSE of 0.081 $m^3/m^3$ and 0.067 $m^3/m^3$ for the ARM and TxSON networks, which is 0.009 $m^3/m^3$ and 0.019 $m^3/m^3$ less than the SMAP L4 SSM dataset, respectively.





**Table 3.** Accuracy of the 3-hour surface soil moisture (SSM) from the SMAP L4 SSM product and the STF_SSM datasets. Values in bold indicate the dataset with better performance for each statistic in each row.

| | SMAP L4 SSM (9 km) | | | | | STF_SSM (1 km) | | | | |
|---|---|---|---|---|---|---|---|---|---|---|
| | CC | RMSE | Bias | ubRMSE | KGE | CC | RMSE | Bias | ubRMSE | KGE |
| ARM | **0.744** | 0.090 | **0.001** | **0.060** | 0.544 | 0.696 | **0.081** | 0.023 | 0.065 | **0.593** |
| CW3E | **0.818** | **0.074** | **-0.008** | **0.046** | **0.398** | 0.777 | 0.097 | -0.047 | 0.051 | 0.216 |
| FLUXNET-AMERIFLUX | **0.920** | **0.060** | **-0.009** | **0.055** | **0.746** | 0.889 | 0.073 | -0.030 | 0.066 | 0.645 |
| SCAN | **0.628** | **0.097** | **-0.011** | **0.065** | **0.305** | 0.582 | 0.097 | -0.016 | 0.066 | 0.304 |
| SNOTEL | **0.512** | **0.111** | 0.021 | **0.084** | **0.189** | 0.457 | 0.114 | 0.033 | 0.086 | 0.136 |
| TxSON | **0.769** | 0.086 | -0.018 | 0.047 | 0.358 | 0.750 | **0.067** | 0.020 | **0.045** | **0.514** |
| USCRN | **0.708** | **0.088** | **-0.012** | **0.055** | **0.357** | 0.674 | 0.097 | -0.032 | 0.057 | 0.295 |
| Mean | **0.728** | **0.086** | -0.005 | **0.059** | **0.414** | 0.689 | 0.090 | -0.007 | 0.062 | 0.386 |

**3.4 SSM data accuracy across land cover types**

Generally, the accuracy of satellite-derived SSM datasets varies between land cover types, because the penetration capacity of remote sensing signals can be affected by land cover types. Therefore, it is necessary to assess the performance of the SSM datasets under different land cover types. According to the NLCD land cover product from 2015 to 2023, we separated the CONUS into nine types, including the developed, barren, forest, shrub, grassland, pasture, crops, wetlands, and changed. The former eight types are the existing categories (e.g., shrub and grassland) or composite categories (e.g., forest is composed of deciduous, evergreen, and mixed forest) in the NLCD product. The "changed" category refers to the areas where land use has changed between 2015 and 2023.

At the 3-hour scale, it is seen from Figures 8a and 8b that the SMAP L4 product has slightly better performance than the generated STF_SSM dataset for most land cover types. As shown in Table 4, the mean CC and RMSE values of the SMAP L4 product are 0.611 and 0.107, which are 0.042 and 0.002 better than those of the STF_SSM dataset, respectively. However, both SSM datasets exhibit lower accuracy in wetlands compared to other land cover types. Meanwhile, the SSM accuracy has a larger variation across forest pixels, shrub pixels, and developed area pixels than the other land cover types. This is primarily because the topsoil in these land cover types is covered by woody plants and man-made features, which influences SSM observations and leads to a loss of accuracy. The highest accuracy is observed in barren (with an RMSE of 0.083 $m^3/m^3$ for the STF_SSM dataset) and grassland areas (with an RMSE





of 0.083 m³/m³ for the SMAP L4 product), as these types tend to have less cover. The land

cover changes observed in the "changed" category also provide satisfactory accuracy. This is

because approximately 50% of the land cover change samples consist of barren or grassland

areas, which contribute to higher SSM accuracy. These patterns are also reflected in the daily

SSM datasets (Figures 8c and 8d). Moreover, the Crop-CASMA dataset (with mean CC and

RMSE values of 0.440 and 0.111, respectively) has a lower performance than the STF_SSM

dataset across all land use covers, highlighting the reliability of the generated STF_SSM dataset

at a 1-km spatial resolution.

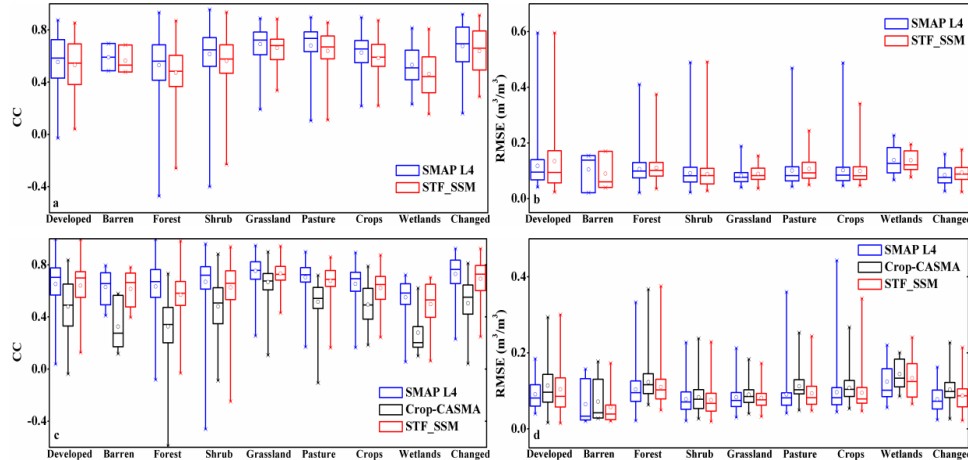


**Figure 8.** Accuracy of the surface soil moisture (SSM) datasets under different land cover types.
The "changed" type refers to areas where land cover type changed between 2015 and 2023. (a)
and (b) are the correlation coefficient (CC) and the root mean square error (RMSE) of SSM
datasets at the 3-hour scale. (c) and (d) are the CC and RMSE of SSM datasets at the daily

scale.







**Table 4.** Mean correlation coefficient (CC) and root mean square error (RMSE) of surface soil moisture (SSM) datasets under different land cover types.

| | 3-hour | | | | Daily | | | | | |
| | SMAP L4 | | STF_SSM | | SMAP L4 | | Crop-CASMA | | STF_SSM | |
| | CC | RMSE | CC | RMSE | CC | RMSE | CC | RMSE | CC | RMSE |
| --- | --- | --- | --- | --- | --- | --- | --- | --- | --- | --- |
| Developed | 0.554 | 0.117 | 0.530 | 0.135 | 0.653 | 0.091 | 0.480 | 0.114 | 0.640 | 0.104 |
| Barren | 0.591 | 0.105 | 0.564 | 0.083 | 0.601 | 0.103 | 0.208 | 0.117 | 0.576 | 0.092 |
| Forest | 0.530 | 0.151 | 0.472 | 0.156 | 0.634 | 0.104 | 0.327 | 0.123 | 0.570 | 0.110 |
| Shrub | 0.615 | 0.091 | 0.562 | 0.089 | 0.669 | 0.078 | 0.481 | 0.083 | 0.627 | 0.076 |
| Grassland | 0.691 | 0.083 | 0.662 | 0.089 | 0.753 | 0.083 | 0.669 | 0.090 | 0.734 | 0.082 |
| Pasture | 0.680 | 0.101 | 0.639 | 0.107 | 0.704 | 0.091 | 0.517 | 0.112 | 0.678 | 0.093 |
| Crops | 0.626 | 0.102 | 0.585 | 0.099 | 0.654 | 0.097 | 0.494 | 0.108 | 0.623 | 0.095 |
| Wetlands | 0.532 | 0.138 | 0.463 | 0.138 | 0.551 | 0.124 | 0.279 | 0.145 | 0.498 | 0.134 |
| Changed | 0.677 | 0.084 | 0.639 | 0.094 | 0.730 | 0.078 | 0.505 | 0.104 | 0.693 | 0.088 |
| Mean | 0.611 | 0.108 | 0.569 | 0.110 | 0.661 | 0.094 | 0.440 | 0.111 | 0.627 | 0.097 |

**3.5 SSM data accuracy across topographic conditions**

As a significant soil-forming factor, terrain is one of the determinants of SSM variations. Particularly, SSM could have strong spatial variability in areas with complex topographic conditions. We analyzed the accuracy of the 1-km Crop-CASMA and STF_SSM datasets under different topographic conditions. As shown in Figures 9a and 9b, both the Crop-CASMA and STF_SSM datasets show a decrease in accuracy (i.e., CC) with increasing elevation. However, the STF_SSM dataset shows a slower decline in accuracy (with a slope of -0.055) compared to the Crop-CASMA dataset (with a slope of -0.100). Under complex terrain conditions (i.e., larger slope), the accuracy of both SSM datasets is reduced. It can be seen from Figures 9c and 9d that the CC of the Crop-CASMA dataset decreases more sharply as the slope increases (with a slope of -0.023) while the CC in the generated STF_SSM dataset declines more gradually (with a slope of -0.011). This suggests that the STF_SSM dataset is more reliable than the Crop-CASMA dataset in complex terrain conditions.

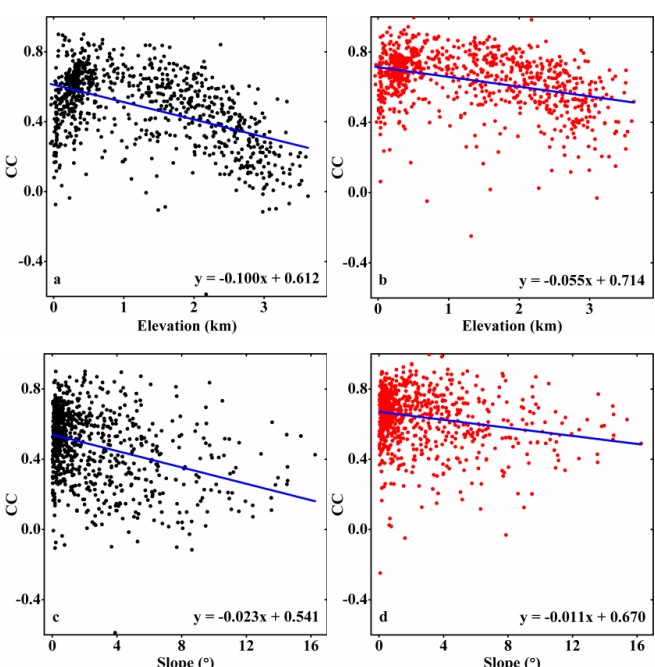


**Figure 9.** Accuracy of the estimated surface soil moisture (SSM) from the 1-km Crop-CASMA and generated STF_SSM datasets along changing topographic conditions, denoted by elevation and slope. (a) and (c) refer to the 1-km Crop-CASMA SSM dataset. (b) and (d) are the generated STF_SSM dataset.



## 4. Discussion

### 4.1 Implications

**Ma et al. (2021)** adopted the daily 25-km, 10-day composited CCI SSM time-series to define agricultural drought events. However, this 10-day composition approach may not capture the rapidly developed drought phenomena (e.g., flash drought). In contrast, the generated 3-hour, 1-km STF_SSM dataset has advantages over the CCI dataset, because the STF_SSM dataset provides more detailed and continuous SSM information in both the temporal and spatial dimensions. This implies that the STF_SSM dataset can detect both long-term and flash drought events at a finer scale.

Figure 10 exhibits the SSM variation under four drought events in Oklahoma (January to February, 2018), Alabama (May to December, 2016), California (January to December, 2020), and Nevada (January to December, 2020) (Time Series | U.S. Drought Monitor, 2024). The SSM in our dataset shows a clear response to the drought event. As shown in Figure 10a, during the Oklahoma's drought in early 2018 **(Shephard et al., 2021)**, SSM is about 0.08 m$^3$/m$^3$ lower than the multi-year average value from 2015 to 2023. The drought was gradually alleviated in February. Alabama's drought in 2016 began around May and continued into December (Figure 10b) **(Noel et al., 2020)**, as the SSM value began to deviate from the average in May and remained in a lower range. In 2020, California and Nevada suffered extreme and long-term droughts **(Williams et al., 2022),** Figures 10c and 10d show that SSM values in the two states were generally less than the average and continued for the entire year.

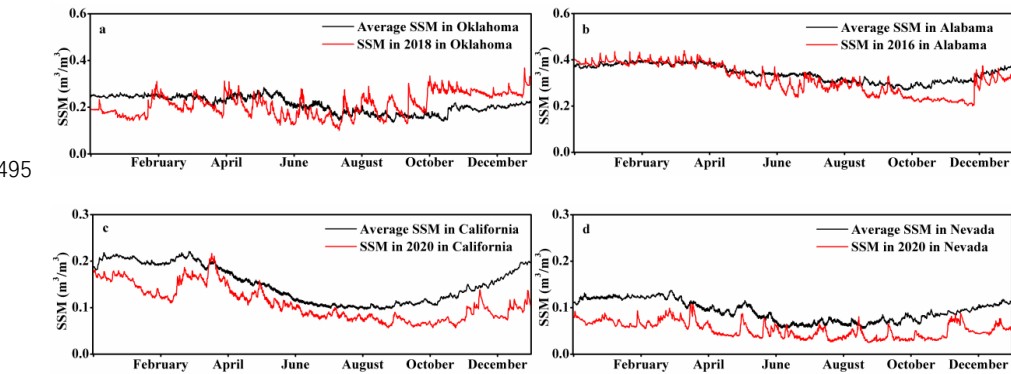

**Figure 10.** Surface soil moisture (SSM) variations under drought events. Black lines refer to the average SSM values calculated from 2015 to 2023. Red lines represent the SSM values for



the corresponding year. (a) the drought in Oklahoma in 2018. (b) the drought in Alabama in
2016. (c) the drought in California in 2020. (b) the drought in Nevada in 2020.

**4.2 Accuracy and latency time for the STF_SSM data**

Drought monitoring needs real-time or near real-time soil moisture data, which requires
high data accuracy and short latency time. Since the strategies of characteristics extraction for
training the STF_SSM dataset from 2015 to 2023 depend on the complete SSM time-series
data over the entire year, it is difficult to directly update the near real-time data using this
strategy. Therefore, we propose three alternative strategies for characteristics extraction to
examine on a randomly selected date in 2023 (2023-01-10). This first alternative is using all
data in 2022; the second alternative is using available data in 2023 (before 2023-01-10); the
third is using all data in 2022 plus available data in 2023. By referring to the corresponding
STF_SSM scenes (predict using all data in 2023), we found that using available data in 2023
has the greatest performance among the three strategies of characteristics extraction, with a
mean CC and RMSE of 0.999 and 0.006 $m^3/m^3$ (Table 5). Meanwhile, the CC and KGE of
using all data in 2022 plus available data in 2023 are 0.964 and 0.947, which are 0.004 and
0.009 higher than those using data only in 2022. This means that using characteristics extracted
from temporally adjacent data in 2023 improves fusion accuracy. According to the investigation
from the official website, the near real-time SMAP L4 SSM product and Crop-CASMA SSM
dataset usually have three and two days of latency, respectively. Thus, the latency time of the
near real-time STF_SSM scene could be approximately three days.

**Table 5.** Accuracy of near real-time STF_SSM data (on 2023-01-10) production using different
strategies of characteristics extraction.

|  | CC | RMSE | Bias | ubRMSE | KGE |
|---|---|---|---|---|---|
| Accuracy using all data in 2022 | 0.960 | 0.035 | -0.010 | 0.033 | 0.938 |
| Accuracy using all data in 2022 plus available data in 2023 | 0.964 | 0.032 | -0.008 | 0.031 | 0.947 |
| Accuracy using available data in 2023 | 0.999 | 0.006 | 0.000 | 0.006 | 0.992 |

**4.3 Uncertainties and future works**

Although the STF_SSM dataset has good performance in representing the fast changes in
soil moisture, two uncertainties in the data generation need to be noted. First, the spatio-





temporal fusion model used in this study is a data-driven method, which depends on the stable and accurate SMAP L4 SSM product and Crop-CASMA SSM dataset. If either of these datasets stops updating or contains significant errors, the generation and accuracy of the STF_SSM dataset will be impacted. Second, many environmental and ecological variables affect the SSM, such as precipitation, vegetation, temperature, evaporation, and terrain. However, these

variables are not fully considered in the STF_SSM production, decreasing the interpretability of the STF_SSM dataset.

Currently, geostationary satellites have a large potential to provide hourly and spatially fine auxiliary variables to produce SSM datasets. Fusing the auxiliary variables from the geostationary satellites for the generation of hourly SSM data is an ongoing work. However,

the extensive data acquisition and necessary preprocessing steps can significantly increase the time cost of data production without leading to a substantial improvement in accuracy. Thus, balancing data accuracy and generation efficiency is necessary for the downscaling of the SSM dataset in the future. Compared with existing SSM datasets, the generated SSM dataset in this study has advantages in terms of spatio-temporal resolution. It is worthwhile to further explore

its potential applications, such as monitoring drought severity and occurrences, quantifying wildfire danger levels, evaluating responses of agriculture and natural ecosystems to soil moisture dynamics, and understanding local and regional hydrological processes.



## 5. Conclusions

In this study, we developed a spatio-temporal fusion model to generate the first spatially seamless 3-hour, 1-km STF_SSM dataset in the CONUS from 2015 to 2023. This dataset integrated the 3-hour, 9-km SMAP L4 SSM product from NASA and the daily, 1-km Crop-CASMA SSM dataset from USDA-NASS. The former provided fine temporal resolution and the latter provided detailed spatial details. To deal with the mismatch between the two datasets

in terms of temporal resolution and spatial coverage, the HCTSA-based time-series mining method was used to extract spatially seamless characteristics from both SSM time-series data for each year. Four characteristics extracted at 1-km and 9-km spatial resolutions (minimum, maximum, mean, and median of SSM time-series) were employed as inputs of the fusion model. By coupling with each 3-hour, 9-km SMAP L4 scene, the downscaled 3-hour, 1-km STF_SSM

scene was simulated, in turn. Data validation at the daily scale showed that the generated 1-km STF_SSM dataset (with a mean CC and ubRMSE of 0.716 and 0.057 $m^3/m^3$) outperforms the 1-km Crop-CASMA SSM dataset (with a mean CC and ubRMSE of 0.584 and 0.072 $m^3/m^3$) when compared to *in-situ* measurements. At the 3-hour scale, the accuracy of the 9-km SMAP L4 SSM product (with a mean CC and ubRMSE of 0.728 and 0.059 $m^3/m^3$) is slightly higher

than that of the 1-km STF_SSM dataset (with a mean CC and ubRMSE of 0.689 and 0.062 $m^3/m^3$). Additionally, the STF_SSM dataset has a better performance than the Crop-CASMA dataset under complex terrain conditions. Overall, the generated 3-hour, 1-km STF_SSM dataset is reliable and has great potential for applications at various spatio-temporal scales. The proposed STF_SSM dataset can be freely acquired from

https://doi.org/10.6084/m9.figshare.28188011.

**Data availability**

The STF_SSM dataset is available at https://doi.org/10.6084/m9.figshare.28188011 (Yang et al., 2025).

**Author contributions**

Conceptualization: HY. Data curation: HY, JY, TEO. Formal analysis: HY, JY, MX. Funding acquisition: JY, TEO, MX. Methodology: HY. Software: HY, JY. Resources: HY, JY, TEO, ESK, MX, CBZ. Supervision: JY, TEO, CBZ. Validation: HY, TEO, ESK. Visualization: HY. Writing – original draft preparation: HY. Writing – review & editing: HY, JY, TEO, ESK, 575 MX, CBZ. All authors have reviewed and approved the manuscript.

**Competing interests**

At least one of the (co-)authors is a member of the editorial board of Earth System Science Data.

**Acknowledgment**

The authors like to thank the NASA NSIDC, USDA-NASS, Global Energy and Water Cycle Experiment (GEWEX), European Space Agency (ESA), Oklahoma Climatological Survey for making the SMAP L4 SSM product, the Crop-CASMA SSM dataset, ISMN data, and *in-situ* data from the freely available Oklahoma Mesonet. The Oklahoma Mesonet is jointly operated by Oklahoma State University and the University of Oklahoma, and continued funding for 585 maintenance of the network is provided by the taxpayers of Oklahoma. Data generation and validation were completed utilizing the High-Performance Computing Center facilities of Oklahoma State University at Stillwater.

**Financial support**

Haoxuan Yang, Jia Yang, Tyson E. Ochsner, Erik S. Krueger and Chris B. Zou were 590 supported by the US Geological Survey South Central Climate Adaptation Science Center grant (G23AC0454), Joint Fire Science Program (23-2-01-9), the US National Science Foundation Oklahoma EPSCoR S3OK project (Grant No. OIA-1946093), the US National



Science Foundation Rural Confluence (No. 2316366). Mengyuan Xu was supported by the Shanghai Rising Star Program supported by the Science and Technology Commission of

Shanghai Municipality (No. 24YF2737600).

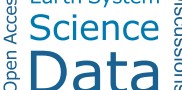

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
