# Peer review of "A 3-hour, 1-km surface soil moisture dataset for the contiguous United States from 2015 to 2023"

_Earth System Science Data, 2025_

## Author Comment (AC1)

The manuscript presents a soil moisture dataset for the contiguous United States with fine temporal and spatial resolution. The authors disaggregate the fine soil moisture data from a daily level to a 3-hour level. I also note that the spatial coverage of soil moisture data is complete. It seems to make the dataset more useful.

Response:

Thank you very much for the positive comments and constructive suggestions. We have carefully considered your comments and have responded as follows. The revised content is marked in red in the response.

I have a concern. In section 4.1, the authors used the drought in four states as a case study to display the decline in soil moisture. However, since drought is always a slow process, it seems the data have potential to characterize changes in drought using the daily soil moisture data. I suggest that the authors supply fast-forming disasters as a case study further to amplify the importance of hourly soil moisture data.

Response:

Thank you for the valuable comment. We agree with the comment and admit that the fast-forming disasters indeed further reflect the advantages of the STF-SSM dataset in terms of fine temporal resolution. We think that flooding is an appropriate case to further illustrate the importance of the SSM dataset. First, there is a correlation between the occurrence of flooding and the change of SSM. Second, the variation of flooding is rapid enough to be observed with the 3-hour SSM data.

Specifically, two flooding events (in South Carolina and Texas) are selected to exhibit the advantages of the developed 3-hour SSM dataset. We found that the SSM variation is sensitive to flooding. Hence, we have revised Section 4.1 and supplied the description.

*Lines 515-529*

*In addition to droughts, SSM is also sensitive to flooding. When a flooding event begins, the SSM value is usually rapidly increased over a short period. To highlight the advantages of the developed 3-hour SSM dataset, we portrayed the SSM variation under two flooding events. Figure 11a shows the flooding in 2015 in Williamsburg, South Carolina, because of extreme precipitation from 2015-10-01 to 2015-10-05. It can be seen that the SSM value in this region began to increase dramatically from the evening of 2015-10-01 to 2015-10-02, and remained at a high level until 2015-10-05. Figure 11b presents the flooding in 2017 in Jefferson, Texas, due to Hurricane Harvey. We found that the SSM value had started to rise on the evening of 2017-08-24 before Hurricane Harvey reached landfall fully (2017-08-25), and peaked on 2017-08-27.*

[Figure]

**Figure 11.** *Surface soil moisture (SSM) variations under flood events. Black lines represent the average SSM values calculated from 2015 to 2023. Red lines are the SSM values for the corresponding year. (a) the flood in Williamsburg, South Carolina in 2015. (b) the flood in Jefferson, Texas in 2017.*

There are also some minor suggestions as follows:

1). The units of RMSE, Bias, and ubRMSE in Tables 2-5 should be provided.

Response:

Thank you for the suggestions. We have supplied the units of RMSE, Bias, and ubRMSE in these tables.

2). Section 2.2: The VIPSTF model includes two different versions. Which version was used in the manuscript? Please explain it.

Response:

We gratefully appreciate your valuable comment. The VIPSTF model contains a spatial weighting- and a spatial unmixing-based version. Considering the difference of accuracy, the selection in this manuscript is the spatial weighting-based version. Also, we have added the corresponding description in the latest manuscript.

*Section 2.2.2*

*In this study, the virtual image pair-based spatio-temporal fusion (VIPSTF) model was employed to generate the 3-hour, 1-km STF_SSM dataset, due to its stable performance, superior computational efficiency, and flexible usage **(Wang et al., 2020; Yang et al., 2023)**. The spatial weighting version of the VIPSTF model was adopted in this study, because of the reliable accuracy.*

3). Line 265: What is the size of the soil moisture dataset? What is the data format? I suggest the authors to provide more detailed information about the STF_SSM dataset.

Response:

We have further added the detailed STF_SSM data format in the manuscript. Specifically, a total of 25567 scenes were produced, and the STF_SSM dataset occupied approximately 1.78 TB of storage space. The Pete supercomputer in Oklahoma state university was employed for data generation, which can provide 6688 processor cores and more than 2 petabytes of storage. Each STF_SSM scene requires approximately 73.0 MB of storage space and takes around 680 seconds to produce. Given the difference in computational efficiency of different computers, we only show

the size of the dataset in the latest manuscript. Thank you for the comments.

*Section 2.2.3*
    *Finally, a total of 25,567 STF  SSM scenes were produced, accounting for approximately 1.78 TB. Each STF_SSM scene requires approximately 73.0 MB of storage space. The Pete High-Performance Computing (HPC) facility at Oklahoma State University was employed for data generation.*

4). Figure 3: Is the scene in the first column an average of all the intraday scenes? Please clarify the specific time for each scene.

Response:
    Thank you for the comments. The temporal resolution of the Crop-CASMA SSM data is daily. Hence, the exhibited Crop-CASMA SSM scene in Figure 3 is not an average of all the intraday scenes. By comparison, the temporal resolution of the SMAP L4 SSM and STF_SSM datasets is 3 hours, so we have exhibited some SSM scenes at special time points. We have described this information in the title of Figure 3.

5). Line 475: The potential for hourly soil moisture data applications needs to be further emphasized. That is the main driver of fine soil moisture dataset development.

Response:
    Thank you for the valuable suggestions. We have selected two flooding events to exhibit the advantages of the developed 3-hour SSM dataset. Since the SSM is always increased for a short period, when a flooding event occurs. Hence, we have revised Section 4.1 and supplied the corresponding contents.

*Lines 505-519*
    *In addition to droughts, SSM is also sensitive to flooding. When a flooding event begins, the SSM value is usually rapidly increased over a short period. To highlight the advantages of the developed 3-hour SSM dataset, we portrayed the SSM variation under two flooding events. Figure 11a shows the flooding in 2015 in Williamsburg, South Carolina, because of extreme precipitation from 2015-10-01 to 2015-10-05. It can be seen that the SSM value in this region began to increase dramatically from the evening of 2015-10-01 to 2015-10-02, and remained at a high level until 2015-10-05. Figure 11b presents the flooding in 2017 in Jefferson, Texas, due to Hurricane Harvey. We found that the SSM value had started to rise on the evening of 2017-08-24 before Hurricane Harvey reached landfall fully (2017-08-25), and peaked on 2017-08-27.*

[Figure]

***Figure 11.*** *Surface soil moisture (SSM) variations under flood events. Black lines represent the average SSM values calculated from 2015 to 2023. Red lines are the SSM values for the corresponding year. (a) the flood in Williamsburg, South Carolina in 2015. (b) the flood in Jefferson, Texas in 2017.*

6). The references listed also provide 1-km soil moisture data or downscaling methods, which may be helpful to your work. In addition, what the difference is between the proposed and listed methods.
https://doi.org/10.1016/j.jag.2023.103572
https://doi.org/10.1016/j.rse.2022.113334
https://doi.org/10.1016/j.rse.2024.114579

Response:

Thank you very much for the suggestions. The mentioned references mainly contain the use of synthetic spatial radar (SAR) data to estimate high-resolution SSM data. The proposed methods mainly consider the optical/thermal-infrared data to predict the fine SSM data. Both methods can effectively estimate SSM and improve its spatial resolution. We admitted that using the SAR data is also reliable and accurate. However, this method is limited by the coarse revisit period and narrow swath width of SAR data, especially for estimating a large, high temporal resolution SSM dataset. We have revised the introduction and supplied this description in the latest manuscript.

*Lines 126-131*

*In addition, synthetic aperture radar (SAR) data are also beneficial for generating high-resolution SSM datasets. Since microwave signals can penetrate the cover of clouds or haze, the SSM estimation can avoid the influence of weather factors. However, producing a large-scale, fine temporal resolution SSM product is limited by the coarse revisit period and narrow swath width of SAR data (**Wang et al., 2023; Zhu et al., 2023; Fan et al., 2025**).*

*Fan, D., Zhao, T., Jiang, X., García-García, A., Schmidt, T., Samaniego, L., Attinger, S., Wu, H., Jiang, Y., Shi, J., Fan, L., Tang, B.-H., Wagner, W., Dorigo, W., Gruber, A., Mattia, F., Balenzano, A., Brocca, L., Jagdhuber, T., Wigneron, J.-P., Montzka, C., and Peng, J.: A Sentinel-1 SAR-based global 1-km resolution soil moisture data product: Algorithm and preliminary assessment, Remote Sensing of Environment, 318, 114579, https://doi.org/10.1016/j.rse.2024.114579, 2025.*

*Wang, Z., Zhao, T., Shi, J., Wang, H., Ji, D., Yao, P., Zheng, J., Zhao, X., and Xu, X.: 1-km soil moisture retrieval using multi-temporal dual-channel SAR data from Sentinel-1 A/B satellites in a semi-arid watershed, Remote Sensing of Environment, 284, 113334, https://doi.org/10.1016/j.rse.2022.113334, 2023.*

*Zhu, L., Wang, H., Zhao, T., Li, W., Li, Y., Tong, C., Deng, X., Yue, H., and Wang, K.: Disaggregation of remote sensing and model-based data for 1 km daily seamless soil moisture, International Journal of Applied Earth Observation and Geoinformation, 125, 103572, https://doi.org/10.1016/j.jag.2023.103572, 2023.*

---

## Author Comment (AC2)

This manuscript produces a 3-hour, 1-km soil moisture (SM) dataset generated using a spatiotemporal fusion approach that integrates the 3-hour, 9-km SMAP L4 soil moisture product with the 1-day, 1-km Crop-CASMA soil moisture data. The resulting dataset is evaluated against in-situ SM observations. Overall, the manuscript is well-structured, and the generated SM dataset holds significant potential for the scientific community. However, several aspects require clarification and further discussion:

Response:

The authors gratefully appreciate your valuable comments and suggestions. We have carefully considered your comments and have responded as follows. The revised content is marked in red in the response.

1. Validation Approach: The methodology leverages the higher-accuracy SMAP L4 product to capture temporal variations while using the lower-accuracy but higher-resolution Crop-CASMA data to retain spatial details. Consequently, the accuracy of the fused product should theoretically be higher than that of Crop-CASMA but lower than SMAP L4.

Response:

We agree with your point. In this manuscript, both the 3-hour and daily validation were expressed in Sections 3.3 and 3.4. It can be seen that the validated results are consistent with the reviewer's view. That is, the accuracy of the simulated STF_SSM dataset is higher than that of the Crop-CASMA but smaller than that of the SMAP L4. Thank you for the comments.

2. Comparative Analysis: The authors compare their product only with SMAP L4 and Crop-CASMA but do not benchmark it against other 1-km resolution datasets or even higher-resolution (30-m) products (DOI: 10.1038/s41597-021-01050-2). Including such comparisons, or at least discussing them, would provide a more comprehensive evaluation of the dataset's performance.

Response:

Thank you for the comment. We have supplied a discussion in Section 4.3 to analyze six high-resolution SSM datasets. Considering the differences in validation methods, spatial and temporal coverages, and statistical metrics, it is not appropriate to quantify them under a standard. Hence, we suggested that other people select according to their requirements, before using the data.

4.3 Analysis of different fine SSM datasets

Currently, some high-resolution SSM datasets have been published and used. We listed six 1-km SSM datasets at a large scale and exhibited the details of these datasets in Table 6, such as the spatial resolution, temporal resolution, and accuracy. Given the differences in validation methods, spatial and temporal coverages, and statistical metrics, etc, it is difficult to harmonize these

*datasets to the same standard to quantify accuracy. Therefore, before using the data, it is necessary to further select the more suitable SSM dataset according to the requirements.*

**Table 6.** *Comparative analysis for six high-resolution SSM datasets. Statistical metrics for accuracy include root mean square error (RMSE), unbiased root mean square error (ubRMSE), and unbiased root mean square deviation(ubRMSD).*

| References | Area | Spatial resolution | Temporal resolution | Accuracy $(m^3/m^3)$ |
|---|---|---|---|---|
| This study | CONUS | 1-km | 3-hour | ubRMSE = 0.057 |
| (Vergopolan et al., 2021) | CONUS | 30-m | 6-hour | RMSE = 0.07 |
| (Fang et al., 2022) | Global | 1-km | daily | ubRMSE = 0.063 |
| (Zheng et al., 2023) | Global | 1-km | daily | ubRMSE = 0.045 |
| (Han et al., 2023) | Global | 1-km | daily | ubRMSE = 0.050 |
| (Song et al., 2022) | China | 1-km | daily | ubRMSD = 0.074 |

*Fang, B., Lakshmi, V., Cosh, M., Liu, P., Bindlish, R., and Jackson, T. J.: A global 1-km downscaled SMAP soil moisture product based on thermal inertia theory, Vadose Zone Journal, 21, e20182, https://doi.org/10.1002/vzj2.20182, 2022.*

*Han, Q., Zeng, Y., Zhang, L., Wang, C., Prikaziuk, E., Niu, Z., and Su, B.: Global long term daily 1 km surface soil moisture dataset with physics informed machine learning, Sci Data, 10, 101, https://doi.org/10.1038/s41597-023-02011-7, 2023.*

*Song, P., Zhang, Y., Guo, J., Shi, J., Zhao, T., and Tong, B.: A 1 km daily surface soil moisture dataset of enhanced coverage under all-weather conditions over China in 2003–2019, Earth Syst. Sci. Data, 14, 2613–2637, https://doi.org/10.5194/essd-14-2613-2022, 2022.*

*Vergopolan, N., Chaney, N. W., Pan, M., Sheffield, J., Beck, H. E., Ferguson, C. R., Torres-Rojas, L., Sadri, S., and Wood, E. F.: SMAP-HydroBlocks, a 30-m satellite-based soil moisture dataset for the conterminous US, Sci Data, 8, 264, https://doi.org/10.1038/s41597-021-01050-2, 2021.*

*Zheng, C., Jia, L., and Zhao, T.: A 21-year dataset (2000–2020) of gap-free global daily surface soil moisture at 1-km grid resolution, Sci Data, 10, 139, https://doi.org/10.1038/s41597-023-01991-w, 2023.*

3.    Temporal Variability Discussion: It is recommended that the authors expand their discussion on the temporal variations of the generated SM dataset within a single day, in addition to the analysis presented in Figures 6 and 7. This would help highlight the advantages of the product in capturing sudden SM changes compared to daily-scale products.

Response:
    To address this problem, we have added a case of sudden change of SSM in Section 4.1, i.e., two flooding events in Texas and South Carolina. According to two flooding events, the SSM time-series have been exhibited in Figure 11. On this basis, we can estimate the duration and occurrence of flooding on a 3-hour scale. In this case, the advantages of the 3-hour SSM dataset can be further amplified.
    In addition, the two SSM time-series within a single day are displayed in Figure 5. It can be seen that the Pixel 2 in Figure 4 was experiencing precipitation, because the SSM values are increased from 1:30 to 7:30. The intra-day variation of SSM is difficult to be observed by daily-scale data.
    Thank you for the valuable suggestions.

*Lines 515-529*

    *In addition to droughts, SSM is also sensitive to flooding. When a flooding event begins, the SSM value is usually rapidly increased over a short period. To highlight the advantages of the developed 3-hour SSM dataset, we portrayed the SSM variation under two flooding events. Figure 11a shows the flooding in 2015 in Williamsburg, South Carolina, because of extreme precipitation from 2015-10-01 to 2015-10-05. It can be seen that the SSM value in this region began to increase dramatically from the evening of 2015-10-01 to 2015-10-02, and remained at a high level until 2015-10-05. Figure 11b presents the flooding in 2017 in Jefferson, Texas, due to Hurricane Harvey. We found that the SSM value had started to rise on the evening of 2017-08-24 before Hurricane Harvey reached landfall fully (2017-08-25), and peaked on 2017-08-27.*

[Figure]

***Figure 11.*** *Surface soil moisture (SSM) variations under flood events. Black lines represent the average SSM values calculated from 2015 to 2023. Red lines are the SSM values for the corresponding year. (a) the flood in Williamsburg, South Carolina in 2015. (b) the flood in Jefferson, Texas in 2017.*

4.      Figure 3: The date and time of the SM data should be explicitly stated in the figure title for clarity.

Response:
      In the updated version, we have revised the title of Figure 3 and supplied the detailed date and time. We appreciate your suggestions.

***Figure 3.*** *Spatial pattern of Surface Soil Moisture (SSM) in the Crop-CASMA SSM dataset (left), SMAP L4 SSM product (middle), and the STF_SSM dataset (right) on 2015-04-01 (01:30), 2017-06-08 (07:30), 2019-08-16 (13:30), and 2021-10-25 (19:30). Both the SMAP L4 and STF_SSM datasets are exhibited at the 3-hour scale, while the Crop-CASMA SSM dataset is displayed at the daily scale. The basemap is from Esri, Earthstar Geographics, and the GIS User Community.*

5.      Figure 8 / Table 4: It is recommended that the authors provide the number of validation sites corresponding to each land cover type, either in Figure 8 or Table 4, to enhance transparency in the validation process.

Response:
      Thank you for the suggestion. The number of validation sites for each land cover type has been added to Table 4. Moreover, we have also revised the title of Figure 8 and supplied the corresponding number for each land cover type.

[Figure]

***Figure 8.*** *Accuracy of the surface soil moisture (SSM) datasets under different land cover types. The "changed" type refers to areas where land cover type changed between 2015 and 2023. (a) and (b) are the correlation coefficient (CC) and the root mean square error (RMSE) of SSM datasets at the 3-hour scale. (c) and (d) are the CC and RMSE of SSM datasets at the daily scale.*

*For each land cover type, the number of validation sites is 45 (developed), 7 (barren), 190 (forest), 211 (shrub), 148 (grassland), 97 (pasture), 69 (crops), 10 (wetlands), and 49 (changed), respectively.*

**Table 4.** Mean correlation coefficient (CC) and root mean square error (RMSE) of surface soil moisture (SSM) datasets under different land cover types.

| Land cover type (number of validation sites) | 3-hour | | | | Daily | | | | | |
|---|---|---|---|---|---|---|---|---|---|---|
| | SMAP L4 | | STF_SSM | | SMAP L4 | | Crop-CASMA | | STF_SSM | |
| | CC | RMSE $(m^3/m^3)$ | CC | RMSE $(m^3/m^3)$ | CC | RMSE $(m^3/m^3)$ | CC | RMSE $(m^3/m^3)$ | CC | RMSE $(m^3/m^3)$ |
| Developed (45) | 0.554 | 0.117 | 0.530 | 0.135 | 0.653 | 0.091 | 0.480 | 0.114 | 0.640 | 0.104 |
| Barren (7) | 0.591 | 0.105 | 0.564 | 0.083 | 0.601 | 0.103 | 0.208 | 0.117 | 0.576 | 0.092 |
| Forest (190) | 0.530 | 0.151 | 0.472 | 0.156 | 0.634 | 0.104 | 0.327 | 0.123 | 0.570 | 0.110 |
| Shrub (211) | 0.615 | 0.091 | 0.562 | 0.089 | 0.669 | 0.078 | 0.481 | 0.083 | 0.627 | 0.076 |
| Grassland (148) | 0.691 | 0.083 | 0.662 | 0.089 | 0.753 | 0.083 | 0.669 | 0.090 | 0.734 | 0.082 |
| Pasture (97) | 0.680 | 0.101 | 0.639 | 0.107 | 0.704 | 0.091 | 0.517 | 0.112 | 0.678 | 0.093 |
| Crops (69) | 0.626 | 0.102 | 0.585 | 0.099 | 0.654 | 0.097 | 0.494 | 0.108 | 0.623 | 0.095 |
| Wetlands (10) | 0.532 | 0.138 | 0.463 | 0.138 | 0.551 | 0.124 | 0.279 | 0.145 | 0.498 | 0.134 |
| Changed (49) | 0.677 | 0.084 | 0.639 | 0.094 | 0.730 | 0.078 | 0.505 | 0.104 | 0.693 | 0.088 |
| Mean | 0.611 | 0.108 | 0.569 | 0.110 | 0.661 | 0.094 | 0.440 | 0.111 | 0.627 | 0.097 |

6.       Figure 9: If feasible, the authors are encouraged to analyze and present the relationship between RMSE, slope, and altitude, as this could provide additional insights into the dataset's accuracy under varying topographic conditions.

Response:
    We appreciate your suggestions to make our manuscript complete. In the latest version, we have modified the content in Section 3.5. Meanwhile, the relationship between RMSE, slope, and elevation has been supplied (Figures 9e to 9h). The results further prove the advantage of the generated STF_SSM dataset.

*3.5 SSM data accuracy across topographic conditions*

*As a significant soil-forming factor, terrain is one of the determinants of SSM variations. Particularly, SSM could have strong spatial variability in areas with complex topographic conditions. We analyzed the accuracy (including CC and RMSE) of the 1-km Crop-CASMA and STF_SSM datasets under different topographic conditions. As shown in Figures 9a and 9b, both the Crop-CASMA and STF_SSM datasets show a decrease in the CC value with increasing elevation. However, the STF_SSM dataset shows a slower decline in accuracy (with a slope of -*

*0.055) compared to the Crop-CASMA dataset (with a slope of -0.100). Under complex terrain conditions (i.e., larger slope), the accuracy of both SSM datasets is reduced. It can be seen from Figures 9c and 9d that the CC of the Crop-CASMA dataset decreases more sharply as the slope increases (with a slope of -0.023) while the CC in the generated STF_SSM dataset declines more gradually (with a slope of -0.011). Likewise, Figures 9e and 9f shows that the RMSE values of both SSM datasets increase with elevation. According to the intercept, the STF_SSM dataset has a slightly greater RMSE than the Crop_CASMA dataset, especially at high altitudes. Meanwhile, with an increase in slope, the STF_SSM dataset has a slower rise in RMSE values than the Crop-CASMA dataset (Figure 9g and 9h). This suggests that the STF_SSM dataset is more reliable than the Crop-CASMA dataset in complex terrain conditions.*

[Figure]

**Figure 9.** *Accuracy of the estimated surface soil moisture (SSM) from the 1-km Crop-CASMA and generated STF_SSM datasets along changing topographic conditions, denoted by elevation and slope. (a), (c), (e) and (g) refer to the 1-km Crop-CASMA SSM dataset. (b), (d), (f) and (h) are the generated STF_SSM dataset.*

---

## Author Comment (AC3)

I see that the authors have addressed several major concerns raised by the other reviewers. The paper is already in good shape and suitable for the journal, with only a few minor comments from my side related to data operations:

Response:
    The authors gratefully appreciate your positive comments and valuable suggestions. We have carefully considered your comments and have responded as follows. The revised content is marked in ==red== in the response.

The availability of an open-access soil moisture dataset with higher spatial and temporal resolution is desirable for both the Ag sector and the Earth system science community. My main concern is the long-term sustainability of the dataset. The current dataset covers the historical period from 2015 to 2023. Will data for the most recent years be made available in the future? The manuscript lacks a discussion on the sustainability and future availability of the dataset, which is crucial for readers who intend to reuse or build upon this work.

Response:
    Thank you for the comments. We agree with the reviewer's view that the sustainability and future availability of the STF_SSM dataset are significant. As a result, we have modified Section 4.2 to illustrate this point further.
    Currently, we have contacted the USGS data repository and transferred the available data. After review, the STF_SSM dataset will also be available on this platform. If this or any other platform requires us to update this dataset in real time, we are willing to do that too. In addition, we have opened the code at https://github.com/hhhhhaoxuanyang/STF_SSM-dataset.git for researchers who need it.

*Lines 553-557*

    *Due to a data-driven approach to production, real-time updates of the STF_SSM data have unavoidable latency time. This is because the latency time of the STF_SSM dataset depends on that of other auxiliary data. According to the investigation from the official website, the near real-time SMAP L4 SSM product and Crop-CASMA SSM dataset usually have three and two days of latency, respectively. Thus, if only available data within the year are adopted to update the STF_SSM data, the latency time of the near real-time STF_SSM scene is at least three days.*

It is good to know the paper used soil moisture data products on Crop-CASMA as inputs and benchmark data. While Crop-CASMA is designed primarily to support USDA NASS operations and is produced operationally during the growing season, it is important to discuss how the new dataset can be used in more real-world applications (like near-real-time crop condition monitoring) beyond historical analysis. Meanwhile, I recommend including a data management plan and ensuring code availability in the manuscript, especially given its submission to a top data journal.

Response:
    Thank you for the comments and suggestions. We admit that the application prospects of the dataset deserve further exploration. Hence, we have revised Section 4.1 and added some potential applications for SSM in agriculture in the future, such as irrigation management and crop yield estimation.
    Furthermore, we are waiting for the review of the dataset by USGS data repository. If it goes well, our dataset will be published in UUSGS data repository for free access. The code has been opened at https://github.com/hhhhhaoxuanyang/STF_SSM-dataset.git.

*Lines 530-536*

*It is clear from the mentioned cases that SSM information is closely linked to drought and flooding. This suggests that SSM can be applied to identify these events and quantify their severity. Thus, the developed STF_SSM dataset has great potential for application, especially in agriculture. For instance, near real-time crop conditions could be observed directly by dynamically monitoring SSM. It will provide a rational basis for refining irrigation management. In addition, SSM information with fine spatio-temporal resolution also has the potential to play an important role in crop yield estimation.*